# Germplasm Resources of Oaks (*Quercus* L.) in China: Utilization and Prospects

**DOI:** 10.3390/biology12010076

**Published:** 2022-12-31

**Authors:** Yong Wang, Chenyu Xu, Qi Wang, Yiren Jiang, Li Qin

**Affiliations:** 1College of Bioscience and Biotechnology, Shenyang Agricultural University, Shenyang 110866, China; 2Insect Resource Research Center for Engineering and Technology of Liaoning Province, Shenyang 110866, China

**Keywords:** oak trees, germplasm resources, distribution in China, utilization

## Abstract

**Simple Summary:**

Oaks (*Quercus* spp.) are a major component of subtropical and temperate forests in the Northern Hemisphere. There are approximately 464 species, and they are dominant tree species in ecosystems. Oaks have a long history of cultivation in Europe, North America, and other continents. They are also cultivated and distributed in most provinces of China. In this paper, *Quercus* germplasm resource distribution, ecological adaptability, and utilization are explored. All evidence indicates that oaks have been widely distributed since ancient times and have greater functional value than other species in the Fagaceae family. This review will provide a deeper understanding of oak conservation, development, and utilization in China.

**Abstract:**

Oaks exhibit unique biological characteristics and high adaptability to complex climatic and soil conditions. They are widely distributed across various regions, spanning 40 degrees latitude and 75 degrees longitude. The total area of oak forest in China is 16.72 million hm^2^. There are 60 lineages of *Quercus* in China, including 49 species, seven varieties, and four subgenera. Archaeological data indicate that oaks were already widely distributed in ancient times, and they are dominant trees in vast regions of China’s forests. In addition, the acorn was an important food for ancestral humans, and it has accompanied human civilization since the early Paleolithic. Diverse oak species are widely distributed and have great functional value, such as for greening, carbon sequestration, industrial and medicinal uses, and insect rearing. Long-term deforestation, fire, diseases, and pests have led to a continuous decline in oak resources. This study discusses the *Quercus* species and their distribution in China, ecological adaptation, and the threats facing the propagation and growth of oaks in a changing world. This will give us a better understanding of *Quercus* resources, and provide guidance on how to protect and better utilize germplasm resources in China. The breeding of new varieties, pest control, and chemical and molecular research also need to be strengthened in future studies.

## 1. Introduction

Oak trees refer to plants in the *Quercus* genus, which is the largest genus (accounting for approximately 50%) in the Fagaceae family [1]. There are approximately 450–500 species of oaks, which are dominant tree species in ecosystems throughout the Northern Hemisphere [2,3,4]. Based on the website Plants of the World Online (https://powo.science.kew.org/) (accessed on 27 December 2022), there are 464 accepted species of *Quercus*. Oak trees exhibit unique biological characteristics and high adaptability to complex climatic and soil conditions [5]. As such, they are widely distributed in tropical, temperate, and cold regions, spanning 40 degrees latitude and 75 degrees longitude [6]. Oaks have a long history of cultivation in Europe and other continents, and oak cultivation is practiced in most provincial regions of China. According to the State Forestry Administration of China, oaks accounts for 13% of natural forest in China. Oaks encompass diverse species and are widely distributed and have greater functional value (such as for greening, soil and water conservation, carbon sequestration, and industrial and medicinal uses) than other species in the Fagaceae family [7,8]. Oak tree classification is challenging, partly due to inter- or intra-specific hybridization and introgression, as a great number of inter- or intra-specific variations exist [9]. Oak leaves vary in size and shape among the different species and likely represent an adaptive response to changing environmental conditions [10,11]. Several research works have studied the influence of climate change on oak growth [12,13,14,15]. Drought impacts leaf phenology, and freeze injury in spring influences young leaves [16,17]. A decrease in Phosphorus (P) availability has been found with global changes [18]. Genomic studies have also investigated patterns of adaptation in European white oaks (*Q. robur*, *Q. petraea*, and *Quercus pubescens*), North American valley oaks (*Q. lobata*), and Asian oaks (*Q. mongolica*) [19,20,21]. Oak genetic studies in China (and all of Asia) are limited relative to those in North America and Europe [22].

The products of oak trees, including acorns, bark, timber, and leaves, have many uses [23,24]. Estimates of the volume of California black oak (*Q. kelloggii*) indicate that approximately 2 billion feet of sawtimber were harvested in commercial timberlands in California in the 1960s. Red and White oaks accounts for 46% of annual hardwood sawn timber (USDA data, 1982) [25]. Sawtooth Oak (*Q. acutissima*) is often used for charcoal production. In Chuzhou city, Anhui province of China, there are 13,300 hm^2^ of firewood forest of *Q. acutissima.* This species has rapid growth, and they are harvested every 6–7 years. Every 667 m^2^ of land can produce 4–5 m^3^ of timber, which yields 1.5 t of charcoal. Every 333 hm^2^ of *Q. acutissima* trees produces approximately 10,000 t of charcoal in Chuzhou. This charcoal is famous in Japan and Korea, earning 1.5 billion yuan per year [26]. In many human societies, oaks are sacred trees and a symbol of strength and endurance. They are often considered full of cultural and historical value. In addition, oak leaves are used for the rearing of tussah (Chinese oak silkworm, *Antheraea pernyi*), an important silk-producing insect in China since approximately the 16th century [27]. The annual production of tussah cocoons has been reported to be approximately 7.10–9.38 × 10^4^ t in China, accounting for 90% of worldwide wild silk production. The edible pupae of tussah is an important insect food, mainly in north China, including in Liaoning, Jilin, Heilongjiang, and Shandong provinces, accounting for 70% of pupal consumption [28].

As the dominant tree species in China, *Quercus* spp. account for 10% of the total forestry area [29]. The importance of *Quercus* in forest composition from ancient to modern times is not common knowledge. Furthermore, the various kinds of oak trees that exist in China and their names have not appeared in other reviews. Here, we review the ancient books (in Chinese), fossil discoveries, and modern basic research on oak trees. The total species and occupied areas are listed and the dominant *Quercus* species are summarized by partial provinces. Archaeological data demonstrate that oaks played an important role in ancient forest construction. Oaks were also used for charcoal materials, timber, and food (acorns). This review will provide a deeper understanding of oak conservation, development, and utilization in China.

## 2. Germplasm Resources and Distribution of Oaks

### 2.1. Germplasm Resources of Oaks

#### 2.1.1. Indigenous Oak Species in China

There are 450–500 species of oaks in the world. Due to their high adaptability to different climatic and soil conditions, oaks are widely distributed in Asia, Europe, North America, and Africa, in the northern hemisphere. Previous studies have reported 66 species of oak trees in China, including 51 species, 14 varieties, and one forma. A more detailed literature review confirmed that there are 67 lineages of *Quercus* in China, including 52 species, 14 varieties, and one forma [8]. After deleting the unaccepted species names or reduplicated synonyms, there are 60 lineages of *Quercus* in China, including 49 species, 7 varieties, and 4 subgenera. (Table 1). Although previous reports categorized *Q. palustris* and *Q. robur* as indigenous oak species in China, other studies have shown that these two species were introduced to China from North America in the early 20th century [30]. Until now, *Q. palustris* and *Q. robur* have been considered foreign-introduced species in most studies, while some reports believe they were transplanted from the Shandong Province and Northeast China. However, no wild trees have been reported in these areas. Recent studies have also classified *Q. sichourensis* and *Q. edithiae* as being in the *Quercus* genus [31]. In North and Northeast China, there are *Q. wutaishanica*, *Q. mongolica*, *Q. dentata*, *Q. aliena* var. *acutiserrata*, *Q. aliena*, and *Q. variabilis*, among other species. These species provide critical habitats for oak silkworms, and approximately 70,000 tons of tussah cocoons are produced each year. In the Henan and Shandong Province, *Q. acutissima*, *Q. variabilis*, *Q. aliena* var. *acutiserrata*, and *Q. chenii* are the main species of oak, with 50,000 tons of tussah cocoons produced each year. In the Southwest of China, including Guizhou, Sichuan, and Shanxi Province, *Q. acutissima*, *Q. variabilis*, *Q. wutaishanica*, and *Q. fabri* are distributed, and approximately 50,000 tons of tussah cocoons are produced annually [29,32].

#### 2.1.2. Imported Oak Tree Species

Although oaks are abundant in China, as a major forest tree species and urban greening species, they have not been extensively studied. In China, the number of tree species available for urban landscaping is quite low. To expand oak tree populations and utilize their resources effectively, China introduced oak trees as greening tree species. The oak trees were imported from Europe and America in the mid-19th century, through foreign missionaries, businessmen, and students [33]. However, oak trees were not systematically introduced or studied in China until in the 1960s, when the Chinese Academy of Forestry introduced more than 10 species of *Quercus* from the United States [34]. This provided abundant materials for the germplasm breeding improvement of oaks in China. Some companies and scientific research institutions have also conducted studies on species introduction. To date, approximately 35 species of *Quercus* have been imported from Europe and America (Table 1). The successful species introduction, domestication, and biological and ecological evaluation of these foreign oaks have been studied [35]. Many of them have been successfully domesticated; for example, *Q. texana* shows good tolerance to waterlogging, and *Q. virginiana* shows high tolerance to drought, salinity, and heavy metal stress [36]. These species have great potential for applications in coastal protection and mining area greening. Meanwhile, several foreign-introduced oaks (e.g., *Q. cinerea* also named *Q. incana*, *Q. cuneata*, and *Q. fraineitto*) were not suitable for the local climate [37].

**Table 1 biology-12-00076-t001:** Germplasm resources of oaks in China.

No.	Species Name	No.	Species Name
1	*Quercus acrodonta* Seemen	49	*Quercus yiwuensis* Huang
2	*Quercus acutissima* Carruth.	50	*Quercus yunnanensis* Franch. (Formal name: *Quercus dentata* subsp. *yunnanensis* (Franch.) Menitsky)
3	*Quercus aliena* Blume	51	*Quercus acutissima* var. *septentrionalis* Liou (Formal name: *Quercus acutissima* subsp. *acutissima*)
4	*Quercus aquifolioides* Rehder. et E.H.Wilson.	52	*Quercus acutissima* var. *depressinucata* H.W.Jen et R.Q.Gao (Formal name: *Quercus acutissima* subsp. *acutissima*)
5	*Quercus baronii* Skan	53	*Quercus aliena* var. *pekingensis* Schottky
6	*Quercus bawanglingensis* Huang, Li et Xing	54	*Quercus aliena* var. *pekingensis* f. *jeholensis* (Liou et Li) H.Wei Jen et L.M.Wang (Formal name: *Quercus aliena* var. *pekingensis* Schottky)
7	*Quercus chenii* Nakai	55	*Quercus aliena* var. *acutiserrata* Maxim.
8	*Quercus cocciferoides* Hand.-Mazz.	56	*Quercus baronii* var. *capillata* (Kozlova) Liou
9	*Quercus dentata* Thunb.	57	*Quercus cocciferoides* var. *taliensis* (A.Camus) Y.C.Hsu et H.Wei Jen
10	*Quercus dolicholepis* A. Camus	58	*Quercus mongolica* var. *crispula* (Blume) H.Ohashi
	*Quercus dolicholepis* var. *elliptica* Y. C. Hsu et H. W. Jen (Formal name: *Quercus dolicholepis* A. Camus)	59	*Quercus mongolica* var. *mongolica*
11	*Quercus edithiae* Skan	59	*Quercus serrata* Murray
12	*Quercus engleriana* Seem.		*Quercus glandulifera* var. *stellatopilosa* W.H.Zhang (Formal name: *Quercus serrata* Murray)
13	*Quercus fabri* Hance		*Quercus serrata* var. *brevipetiolata* (A.DC.) Nakai (Formal name: *Quercus serrata* Murray)
14	*Quercus × fangshanensis* Liou		*Quercus serrata* var. *tomentosa* (B.C.Ding et T.B.Chao) Y.C.Hsu et H.Wei Jen (Formal name: *Quercus serrata* Murray)
15	*Quercus × fenchengensis* H. W. Jen et L. M. Wang	60	*Quercus senescens* var. *muliensis* (Hu) Y.C.Hsu et H.Wei Jen
16	*Quercus franchetii* Skan	61	*Quercus palustris* Münchh. [33]
17	*Quercus fimbriata* Y.C.Hsu et H.Wei Jen	62	*Quercus robur* L. [34]
18	*Quercus gilliana* Rehder. et E.H.Wilson.	63	*Quercus suber* L. [30]
19	*Quercus griffithii* Hook. f. et Thomson ex Miq.	64	*Quercus texana* Buckley [33]
20	*Quercus guyavifolia* H. Lév.	65	*Quercus shumardii* Buckley [37]
21	*Quercus × hopeiensis* Liou	66	*Quercus nigra* L. [36,38]
22	*Quercus kingiana* Craib	67	*Quercus phellos* L. [33,34]
23	*Quercus kongshanensis* Y.C.Hsu et H.W.Jen	68	*Quercus virginiana* Mill. [33,37]
24	*Quercus lanceolata* M.Martens et Galeotti ex A.DC.	69	*Quercus coccinea* Münchh. [34]
25	*Quercus lodicosa* O.E.Warb. et E.F.Warb.	70	*Quercus rubra* L. [34]
26	*Quercus longispica* (Hand.-Mazz.) A.Camus	71	*Quercus falcata* Michx. [33]
27	*Quercus malacotricha* A.Camus	72	*Quercus petraea* subsp. *Petraea* [33]
28	*Quercus marlipoensis* Hu et W.C.Cheng	73	*Quercus velutina* Lam. [34,38]
29	*Quercus mongolica* Fisch. ex Ledeb.	74	*Quercus stellata* Wangenh. [38]
30	*Quercus × mongolicodentata* Nakai	75	*Quercus macrocarpa* Michx. [38]
31	*Quercus monimotricha* Hand.-Mazz.	76	*Quercus alba* L. [38]
32	*Quercus monnula* Y.C.Hsu et H.Wei Jen	77	*Quercus laurifolia* Michx. [36]
33	*Quercus oxyphylla* (E.H.Wilson) Hand.-Mazz.	78	*Quercus* × *schuettei* Trel. [39]
34	*Quercus pannosa* Hand.-Mazz.	79	*Quercus michauxii* Nutt. [39]
35	*Quercus phillyraeoides* A. Gray	80	*Quercus lyrata* Walter [39]
36	*Quercus pseudosemecarpifolia* A. Camus	81	*Quercus ithaburensis* subsp. *macrolepis* (Kotschy) Hedge et Yalt. [26]
37	*Quercus rehderiana* Hand.-Mazz.	82	*Quercus bicolor* Willd. [40]
38	*Quercus semecarpifolia* Sm.	83	*Quercus cerris* L. [40]
39	*Quercus senescens* Hand.-Mazz.	84	*Quercus ellipsoidalis* E.J.Hill. [40]
40	*Quercus setulosa* Hickel et A.Camus	85	*Quercus gambellii* [40]
41	*Quercus sichourensis* (Y.C.Hsu) C.C.Huang et Y.T.Chang	86	*Quercus glauca* [40]
42	*Quercus spinosa* David	87	*Quercus imbricaria* [40]
43	*Quercus tungmaiensis* Y.T.Chang	88	*Quercus libani* [40]
44	*Quercus dentata* subsp. *stewardii* (Rehder) A.Camus	89	*Quercus muehlenbergii* [40]
45	*Quercus tarokoensis* Hayata	90	*Quercus petaea* [40]
46	*Quercus utilis* Hu et Cheng	91	*Quercus prinus* L. [40]
47	*Quercus variabilis* Blume	92	*Quercus salicina* Blume [40]
	*Quercus variabilis* var. *pyramidalis* T.B.Chao, Z.I.Chang et W.C.Li (Formal name: *Quercus variabilis* Blume)	93	*Quercus velutina* [40]
48	*Quercus wutaishanica* Mayr	94	*Quercus stellata* var. *margaretta* [40]

Note: 1–60 represent the native oak tree species in China; 61–94 represent the oak tree species introduced from abroad. The species names in former studies are modified as accepted names.

### 2.2. Distribution of Oak Trees

#### 2.2.1. Historical Distribution of Oaks in China

Oaks are widely distributed in Asia, Europe, North America, and Africa, and are particularly abundant in China, the United States, Russia, and India. Fossils of oaks are also widely distributed throughout geological history, and they are a dominant plant group in strata in the Northern Hemisphere from the Eocene epoch [41]. Archaeological data indicate that oaks were already widely distributed in China in ancient times. The Eocene-epoch *Q. rhombifolia* fossil found in Fushun, Liaoning Province, is the earliest fossil record of oak tree leaves in China. *Quercus* sect. *Heterobalanus* (Oerst.) species are montane sclerophyllous oaks in the Hengduan Mountains and have high tolerance to low temperatures, drought, soil impoverishment, and strong ultraviolet radiation. They are the dominant species, and their fossil record extends back to the Miocene. Early fossils of this sclerophyllous oak were found in Xigaze, the Tibet Autonomous Region (late Miocene), which was named *Quercus tibetensis* H. Xu, T. Su et Z.K. Zhou sp. nov. [42]. Other Sect. Brachylepides oak fossils were found in Xiaolongtan Basin, Yunnan province [43]. The earliest fossil of a deciduous oak was found in the Miocene epoch flora in Dunhua, Jilin province. This origin is later than that of the evergreen oaks. From archaeological research, the origin of *Quercus* may have been in the early stages of the Palaeocene, followed by accelerated species differentiation in the Eocene or Oligocene in East Asia, Europe, and North America. *Quercus praedelavayi* Xing Y.W. et Zhou Z.K. sp. nov. s from the upper Miocene was found in southwestern China [44]. *Quercus heqingensis* n. sp. from the late Pliocene was found in Heqing, Yunan province, China [45].

Deciduous broad-leaved tree species (especially *Quercus*) were dominant in vast regions of China. This was confirmed by archeological discoveries, including the 7000-year-old Xinle site in Shenyang, and the Chahai site in Fuxin County, Liaoning [46]. The analysis of charcoal fragments excavated from the Xiajiadian site (3500–4000 years ago) in Chifeng, Inner Mongolia, showed that the loess hills had a relatively warm and humid climate at the time, and that the zonal vegetation consisted of *Q. mongolica* and *Pinus tabulaeformis* forests [47]. Starch grains found on the surface of stone tools from the Shangzhai site (7000 years ago) in Pinggu, Beijing, show that the North China Plain was inhabited by deciduous broad-leaved zonal vegetation, consisting of oak species (such as *Q. mongolica*, *Q. aliena*, and *Q. acutissima*). The charcoal remains from the Beiqian site in Jimo, Shandong, showed that *Quercus* plants (especially *Q. acutissima*) have been the dominant species in the Jiaodong Peninsula since the Beixin cultural period (7000 years ago) [46].

#### 2.2.2. Current Distribution of Oaks in China

Deciduous oaks are widely distributed in China, and form narrow belts in northern areas; in contrast, evergreen oaks are moderately distributed in southern China [48]. Deciduous oaks are dominant and constructive species (the main species in forest construction) in deciduous broad-leaved forests and mixed broadleaf-conifer forests in temperate zone and warm temperate regions, especially in North China. Their altitudinal distribution ranges from a few meters to 3500 m above sea level. China has three regions with concentrated distributions of oaks: (1) the Liaodong and Jiaodong peninsulas are hilly areas inhabited by deciduous oaks (mainly *Q. mongolica*, *Q. wutaishanica*, and *Q. acutissima*); (2) the Funiu and Dabie mountain areas show the highest diversity of *Quercus* species in China (18 species in total, including almost all species found in eastern, western, southern, and northern China); and (3) a wide range of mountainous areas in Sichuan, Yunnan, and Guizhou, which are inhabited by several unique species of oak trees (18 species known to date) [8] (Figure 1).

The total oak forest area in China is 16.72 million hm^2^, based on the 8th National Forest Resources Survey. Most of these are natural forests, and the area of oak trees is 16.1 million hm^2^. The area of artificial oak forest is 0.61 million hm^2^. The land area occupied by oak tree forests exceeds 100,000 hm^2^ in 17 provincial regions, 500,000 hm^2^ in 10 regions, and 1000,000 hm^2^ in five regions (including Heilongjiang, Jilin, Liaoning, Hebei, and Inner Mongolia) [8]. Oak tree forests account for the highest proportion of dominant tree forests. The top ten tree species are oak, fir, larch, birch, poplar, masson pine, eucalyptus, spruce, Yunnan pine, and cypress, which account for 46.3% of the total forest area in China. There are about 20 species of deciduous oaks, being the main dominant species in temperate zones with broadleaved deciduous forest and mixed broadleaf-conifer forest. *Q. acutissima*, *Q. variabilis*, and *Q. dentata* are widely distributed in Northeast, Southeast, and Southern China. *Q. wutaishanica* and *Q. mongolica*, as representatives of deciduous oaks, are widely distributed in Northeast and North China and also located in Sichuan and Hubei provinces. In Yunnan Province, broadleaf oak timber reserves have reached 0.15 billion m^3^, which represents 43% of broadleaf tree timber reserves. *Q. mongolica* is mostly distributed in Northeast China and east Inner Mongolia. There are 411,000 hm^2^ oak forests in Jilin Province, which accounts for 7% of the local forest area. In Liaoning Province, oak forests account for 43.5% of the local forest area, which is 1.06 million hm^2^. In Hebei Province, oak forests total 0.9 million hm^2^ [49]. Based on the newest National Forest Resources Survey in 2019, *Q. mongolica* was the fifth most important tree (based on numbers), representing 8.294 billion trees and 0.583 billion m^3^ timber reserves. *Q. wutaishanica*, in the thirteenth most important position, has 2.647 billion trees and 0.183 billion m^3^ timber reserves.

### 2.3. Ecological Adaptability of Oak Trees

#### 2.3.1. Morphological and Physiological Adaptability of Oaks

Oak trees vary in their leaf size and shape (Figure 2). This variation might reflect adaptations or plastic responses to different environments [50,51,52]. Leaf variation is influenced by genetic and environmental factors [53]. Oaks are deep-rooted plants. Their root systems are well-developed and deeply distributed, and the main roots of one-year-old trees can reach a soil depth of up to 100 cm [32]. In *Q. variabilis*, the main root length of young seedlings is 10 cm on the 58th day (no fibrous roots) and 50 cm on the 73rd day (with a large number of fibrous roots). The main roots of mature *Q. variabilis* can reach 6–7 m deep [8]. The well-developed root systems of oaks are important for soil and water conservation in mountainous areas. Roots also have varying degrees of plasticity, to adapt to environmental stress based on morphological or physiological plasticity and root chemical changes [54]. In addition, oak trees are obligate mycorrhizal symbionts. Their root hairs are 100–150 μm long and develop an ectomycorrhizal symbiosis with certain fungi. The mycelia of mycorrhizal fungi surround the root hair. They enter the cortex and invade the intercellular spaces. The mycelia extend outwards on the root surface in the form of villi and absorb water and nutrients from the soil, to maintain the growth of the oak [55]. Annual seedlings inoculated with mycorrhizal fungi can attain approximately double the biomass of control groups. For example, *Q. wutaishanica* inoculated with symbiotic fungi (such as *Comphidius viscidus* and *Russula foeten*) showed improved seedling growth, net photosynthetic rate, and total nitrogen and phosphorus contents [56]. Mycorrhizae affect the absorption of phosphorus fertilizers and have an antibiotic effect, thus protecting the roots of oaks from infestation by root rot fungi [8]. Mycelia can also secrete various types of extracellular enzymes that promote the decomposition of organic matter in the soil [57,58]. Based on these functions, mycorrhizal fungi can be used for the seedling culture and repopulation of oak trees.

Most oaks are heliophilous or neutral species that do not have strict soil condition requirements for growth. They grow rapidly in moist, fertile, and well-drained neutral or slightly acidic sandy loam soil (particularly in ravines and foothills). They can also tolerate stress and soil infertility, because of their deep-rootedness and mycorrhizal formations. In addition, acorns have a strong sprouting ability, while trees are not transplantation-resistant. Oaks are adaptable to a wide range of temperatures. Deciduous species of oak trees (such as *Q. mongolica*, *Q. wutaishanica*, and *Q. dentata*) are quite cold-resistant, whereas evergreen species are relatively demanding, requiring warmer temperature conditions [59]. Deciduous species are also relatively drought-resistant, whereas evergreen species are moisture-loving. The drought-tolerant species of oaks indigenous to China include *Q. acutissima*, *Q. variabilis*, *Q. wutaishanica*, and *Q. mongolica*. Moisture-loving species in southern China include *Q. fabri*, *Q. serrata*, and *Q. chenii*. Most foreign-introduced species of oaks are hygrocolous and highly moisture-resistant. In general, oaks exhibit strong resistance to wind, fire, pollution, and smoke. Species such as *Q. variabilis*, *Q. acutissima*, *Q. wutaishanica*, *Q. suber*, and *Q. mongolica* have a thick bark and exhibit fire resistance, which makes them ideal fire-resistant tree species [60]. Owing to its resistance to smoke and poisonous gas, *Q. mongolica* is a dominant tree species in the greening and isolation belts of industrial and mining areas [61]. Some species of oaks also exhibit saline-alkaline tolerance, such as *Q. texana* and *Q. nigra* [62]. In addition, *Q. variabilis* can absorb and accumulate heavy metals in suburban areas [63].

#### 2.3.2. Climate Change Influences

Oak is one of the most diverse and ecologically important trees in the Northern hemisphere. They exhibit high tolerance to different environments and have proved useful in evolutionary mechanism research [64,65]. Climate change effects the oak pollen season, especially the start dates and season lengths [66]. In the Mediterranean basin, *Q. pubescens*, as an downy oak, is often used for anti-drought research in morphoanatomy, physiology, and genetic evolution. A significantly earlier senescence increased the sugar content in leaves, to maintain a higher photosynthetic potential [67]. Drought induced an increase in oxidative pressure from the transcript level [68]. Under a constant CO_2_ concentration, the net primary production was positively affected by longer vegetation periods and negatively by respiration costs in European oak trees [69]. In the Mediterranean region, climate change induces heat waves and droughts, disturbing forest species and affecting productivity. The cork oak (*Q. suber*) is resilient and cork growth rapidly recovers when droughts finish [70]. Under long-term environmental changes, trees mainly rely on phenotypic changes. In two oaks, *Q. pyrenaica* (more tolerant to drought) and *Q. petraea* (less tolerant to drought), *Q. petraea* displayed a greater response to moisture availability, by triggering a tighter stomatal control across genetic groups [71].

Oaks comprise 13% of the natural forest in China. Studies specifically addressing oaks adaptation to climate change in China are needed. Greenhouse gases will affect the species geographical distribution and change the richness distribution pattern. Based on 35 oak species and data of 19 bioclimatic variables in China, the *Quercus* distribution will migrate to high altitudes or high latitudes, from being primarily distributed in the area of southwestern China. A high percentage of species loss will happen in mountainous areas, while other regions will gain species due to a northward shift in the years 2050 and 2070 [72].

### 2.4. Threats to Oak Trees

There are several diseases that can influence oak trees, including leaf, trunk, root, and acorn diseases. Leaf diseases are the most common, and most pathogens are fungi. The infected leaves exhibit growth deficiencies and death. Powdery mildew [73], brown leaf spot, rust disease, and “frog eye” disease mainly affect healthy leaves [74] (Figure 3). White rot disease is found in the trunk and roots.

There were 624 kinds of pests reported in 2010 as damaging the leaves, trunks, and acorns in oak trees [75]. Regarding leaf pests, *Malacosoma neustria*, *Camptoloma interiorata*, *Parasa consocia*, *Phalera assimilis*, *Phalerodonta albibasis*, and *Fentonia ocypete* greatly impact leaf growth. In recent years, an emerging oak pest, *Rhynchaenus maculosus*, has caused spectacular damage in Jilin, Heilongjiang, and Liaoning Province. The morphology, life cycle, and biology of this pest were studied in our laboratory. This pest had no reports till 2012 from when it was first recorded as a new species in China in 1987. It can induce leaf damage symptoms, including blister-like blotches on leaf margins. This pest is an univoltine insect and overwinters as an adult in the leaf litter. Both the larvae and adults can influence leaf growth. The leaf damage in Liaoning province increased from 6.9% in 2016 to 15.4% in 2018 [76,77]. Common trunk and acorn pests include *Mallambyx raddei*, *Laspeyresia splendana*, and *Curculio arakawai* (Figure 3).

Other than disease, the illegal timber trade and harvesting for charcoal also threaten oak resources. Deforestation for farming and increasing mountain fires influence oak growth. For example, *Quercus variabilis* bark is widely used for cork production. Based on the red list statistics, these are 32 critically endangered species and 57 endangered species of *Quercus* (https://www.iucnredlist.org/) (accessed on 27 November 2022). In China, there are 12 species of oak trees whose numbers are decreasing, including *Q. chenii*, *Q. edithiae*, *Q. fimbriata*, *Q. kingiana*, *Q. lodicosa*, *Q. marlipoensis*, *Q. mongolica*, *Q. sichourensis*, *Q. utilis*, *Q. macrocarpa*, *Q. alba*, and *Q. libani*. Two of them, *Q. fimbriata* and *Q. marlipoensis*, are listed as critically endangered species. Four of these, *Q. edithiae*, *Q. kingiana*, *Q. lodicosa*, and *Q. utilis*, are listed as endangered species (https://www.iucnredlist.org/) (accessed on 27 November 2022).

## 3. Utilization of Oak Trees

The utilization of oak trees has a long history in China. Oak timber was used as charcoal and a building material, and acorns were an important source of food for ancient people. The Dadiwan site (7300–7800 years ago), Yangshao site (25,000 years ago), and the Qijia cultural period (approximately 4000 years ago) show the presence of oaks in the surroundings of the sites. Around the Erlitou site (3500–3800 years ago) in Luoyang city, there are many *Q. acutissima* and *Q. aliena*, accounting for 81.8% of the biomass in the local forest, and they are used for charcoal [78]. The above archaeological findings indicate that oaks were an important component of the vegetation in the Yellow River Basin in ancient times.

Acorn was an important food for ancestral humans. In the Peiligang site (8500~7000 years ago), 1800 starch granules were found in 15 stone mills, and most of them were acorn starch [79]. In the Shangshan site, Xiaohuangshan site, and Hemudu site of Zhejiang province, and the Songze site in Shanghai, acorns and oak trees were found. In the Kuahuqiao site in Xiaoshan city and the Hemudu site in Yuyao city of Zhejiang province, a whole pit of stored acorns was found. The charcoal of *Q. acutissima* was also found at the Diaolongbei site (6200 years ago) in Zaoyang, Hubei province, confirming previous findings and indicating that the Yangtze River Basin had a warm and humid climate in remote antiquity [80].

*Quercus* spp. were widely used in different places and periods. They had various Chinese popular names, which were used in different places and periods, including Li, Xiang, Hu, Xiangdou, Liqiu, Zuoli, Zuozi, Xiangshi, Xiangzi, Xu, Zhu, and Zaodou [81]. Indeed, oak trees were used as charcoal materials and as timber. In modern times, oaks are wildly used in energy, edible fungus cultivation, insect rearing, and industrial and food production (Figure 4).

### 3.1. Oak Trees as Sources of Timber

Oak wood has excellent mechanical properties, including high levels of hardness, beautiful wood grains, corrosion resistance, and high resilience. These timbers are widely used for the construction of buildings, bridges, furniture, and textile equipment [82]. Some oak woods are used as raw materials for gun stocks, planer stocks, and tool handles. Oak species with high timber quality include *Q. wutaishanica*, *Q. variabilis*, *Q. aliena*, *Q. mongolica*, *Q. fabri*, *Q. serrata*, *Q. aliena*, *Q. acutissima*, *Q. chenii*, and *Q. stewardii* [83]. The wood densities of *Q. wutaishanica*, *Q. aliena*, *Q. chenii*, and *Q. stewardii* can exceed 0.8 g/cm^3^, and the modulus of rupture of oak trees exceeds 1201 kg/cm^2^ [83]. Based on these qualities, oak wood is generally considered a high-strength material. With the exception of *Q. acutissima* and *Q. aliena*, all species of oaks have a longitudinal compressive strength of at least 560 kg/cm^2^ [84]. In most oak species, the tensile strength parallel to the grain exceeds 1500 kg/cm^2^, especially in *Q. chenii*, where it exceeds 2000 kg/cm^2^. The wood of *Q. wutaishanica*, *Q. aliena*, *Q. fabri*, *Q. chenii*, and *Q. stewardii* is classified as having a high Brinell hardness [85].

### 3.2. Food and Feed Value of Oaks

The annual acorn resources in China total 6–7 million tons. Acorns are rich in nutrients and have a starch content of 50–70%, which makes them an important starch resource [24]. Moreover, acorns are the most important wild woody food resource in China, and have been a major source of food since prehistoric times [86].

The leaves of oak trees are rich in nutrients, vitamins, and minerals, and contain 1.84–5.73% crude protein, 0.23–0.42% crude fat, 1.04–3.87% crude polysaccharides, 9.50–18.67% crude fiber, and 0.34–1.27% total flavonoids [87]. Oaks, especially *Q. dentata* with its broad leaves, are used for a traditional Manchu delicacy, a pancake that is prevalent in Northeast China and northeastern Hebei province. The cake is wrapped in the tender leaves of *Q. dentata* and has the special fragrance of the leaves. Oak timber is also used to cultivate edible mushrooms. As mentioned above, oak trees are the main host plants for tussah larvae. At present, tussahs are mainly fed the leaves of *Q. wutaishanica*, *Q. acutissima*, *Q. mongolica*, and *Q. variabilis*. Most (accounting for 70%) are in Northeast China. Approximately 5000 t of tussahs are produced in Henan and Shandong, fed by *Q. acutissima*, *Q. variabilis*, *Q. aliena*, and *Q. chenii.* Approximately 2500 t of tussahs are produced in Guizhou, Sichuan, and Shanxi, fed by *Q. acutissima*, *Q. variabilis*, *Q. wutaishanica*, and *Q. fabri* [88].

### 3.3. Medicinal Value and Chemical Extracts of Oaks

The bark, leaves, acorns, and acorn shells of oaks can be used for medicinal purposes. Their bark is effective in relieving dampness, reducing heat, and detoxification, and can be used to treat enteritis, diarrhea, dysentery, jaundice, and hemorrhoids (Jilin Chinese Herbal Medicine). Purpurogallin, a compound from nutgalls and oak bark, has antioxidant, anticancer, and anti-inflammatory effects [89]. The leaves can be used to treat bacterial dysentery, pediatric dyspepsia, painful swelling, and hemorrhoids (Heilongjiang Manual of Common Chinese Herbal Medicine). Acorn shells have astringent and hemostatic properties and can be used to treat proctoptosis in diarrhea and dysentery, intestinal bleeding, metrorrhagia, and leukorrhagia. Acorns can relieve diarrhea, due to their astringent properties, and can be used to treat dysentery and hemorrhoidal bleeding (Tang Materia Medica and the Herbal Medicine of Rihuazi).

A clinical study used a decoction of oak tree bark to treat 48 patients with infectious diarrhea and the response rate was 93.75%, indicating a significant therapeutic effect. Aqueous extracts from *Q. mongolica* leaves reduced histopathological changes in type II diabetic mouse models [90]. Modern studies have shown that oak trees are rich in diverse functionally active substances. A review summarized the phytoconstituents from oak extracts and their biological applicability from data of PubMed, Scopus, and Science Direct (2010–2020). Some chemicals can be used as antioxidant, antimicrobial, antiobesity, antidiabetic, and anticancer agents, etc. The most studied oaks, including *Q. brantii*, *Q. infectoria*, and *Q. robur*, have had their biomolecules and biological activity identified. The main bioactive phytochemicals include phenolic compounds, sterols, volatile organic compounds, aliphatic alcohols, and fatty acids [91]. A study revealed that the bark of *Q. mongolica* contained 7-xylosyl-10-deacetyltaxol, 10-deacetyltaxol, and 7-epi-10-deacetyltaxoll [92], and can be used as a raw material for the synthesis of paclitaxel. Subsequently, 23 and 27 volatile chemical compounds were identified in the bark of *Q. wutaishanica* and *Q. acutissima*, respectively [93].

In *Q. wutaishanica* bark, the components with over 1% relative content include hexadecanoic acid 2,3-dihydroxypr, 1-eicosanol, heptacosane, (all-E)-2,6,10,15,19, 23-hexamethyl-2,6,10,14,18,22-tetracosahexaene, Nnonacosane, dibutyl phthalate, eicosane, (Z)-9-octadecenamide, diisooctyl adipate, pentacosane, undecane, butylated hydroxytoluene, and hexadecanoic acid methylester. In *Q. acutissima* bark, the components with over 1% relative content include cyanoacetic acid 1,1-dimethylethyl ester, phytol, (Z,Z)-9,12-octadecadienoic acid, octadecanoic acid, hexadecanamide, (Z)-9-octadecenamide, dibutyl phthalate, hexanedioic acid bis(2-ethylhexyl) ester, 1-docosene, and squalene. The leaves are also rich in functionally active chemicals, and 29 and 28 volatile chemicals were identified in the leaves of *Q. wutaishanica* and *Q. acutissima*, respectively [93]. In the leaves of both species, the components with over 1% relative content include 2-Pentanone,4-hydroxy-4-methyl, dibutyl phthalate, phytol, hexanedioic acid, bis(2-ethylhexyl) ester, and 9-Octadecenamide,(Z)-. A total of 22 chemical compounds (including lupane-3-ketone, friedelin, β-sitosterol, 28-hydroxy-friedelin, and kaempferol) were isolated and identified from the petroleum ether and ethyl acetate extracts of *Q. mongolica* leaves [94]. These leaf extracts were shown to effectively inhibit food-borne bacteria and preserve the freshness of fruits and vegetables [95]. In addition, these leaf extracts improved insulin resistance and reduced blood glucose levels in mice, reducing liver damage [90]. Compared with other research, these chemical extracts are still limited in their exploration and utilization.

Oak trees can also be used to make wood vinegar. Wood vinegar made from *Q. mongolica* had a pH of 4.06, an organic acid mass fraction of 5.20%, a density of 1.0260 g/cm^3^, and a refined yield of 90.7% [96]. This wood vinegar was shown to inhibit *Staphylococcus aureus* and *Escherichia coli*, and the antibacterial components are presumed to be guaiacol and phenol [97]. An extracted solution of oak bark can be used to treat black spots and body ulcers in fish, as well as for burn injuries, empyrosis, trauma, animal and insect bites, and acne in humans. This extract also has significant curative activity in inflammatory, infectious, and suppurative diseases of the human skin and has been shown to be safe and non-genotoxic [98].

### 3.4. Industrial Value of Oak Trees

Oak is one of main raw materials for producing tannin extract, which is an important industrial raw material with diverse applications and a high economic value. The acorn cups, bark, and leaves of *Q. variabilis* can be used to produce tannin extracts with a tannin content of 15.7–15.9%, 8.7–9.3%, and 6.6–7.2%, respectively [99]. Oak trees, including *Q. variabilis* and *Q. suber*, are raw materials for the cork industry. Corks are applied in many industrial fields, because of their light weight, high elasticity, hydrophobicity, chemical stability, and superb physical properties [100]. Oak trees also act as high-quality wood resources for developing biomass energy. Acorn starch (accounting for 50–60% of the total weight) can be made into fuel alcohol, the branches and leaves of oak trees can be fermented to produce methane fuel, and the timber can be processed into solid fuel [101]. Oak species such as *Q. phillyraeoides*, *Q. glauca*, and *Q. acutissima* are used as raw materials for producing fine charcoal, and oak barrels are widely used to store wine [102].

### 3.5. Greening and Ornamental Value of Oaks

Oak trees are characterized by a strong vitality, longevity, a tall form, lush canopy, and broad and colorful leaves. Despite their great value for greening and as ornamental tree species, oak trees have not been widely used in urban or rural areas in China [103]. Poplar trees were commonly used for urban landscaping in the past; however, their pollen can cause various allergic diseases that severely impact the lives of residents. At present, oak trees are receiving attention as a suitable tree species for urban landscaping. European and American countries can provide a reference for urban landscaping uses of oak tree species [104]. *Q. suber*, as an evergreen tree, is the basis of an ecological system, contributing to the survival of many native species and also preventing desertification in vulnerable areas. Many cork forests are recognized as protected ecosystems [105]. Besides this ecological importance, the cork oak has the ability to produce a continuous and renewable cork layer. These cork woods are widely used in building, industry, and wine corks, due to fine physical and chemical properties [106]. Chinese cork oak (*Q. variabilis*) is also an ecologically and economically important deciduous tree species [107]. Together with other East Asian oak species, *Q. acutissima* and *Q. chenii*, they are used as indicator species for local forest health [108].

## 4. Strategies for the Utilization of Oaks

### 4.1. Collecting, Protecting, and Utilizing the Germplasm Resources of Oaks

Although oaks are abundant and have a long history of cultivation in China, most trees currently exist in wild forest, and their utilization is not diversified. At present, oak trees are primarily used to make timber, produce charcoal, raise tussahs, and cultivate edible mushrooms. Long-standing deforestation has led to a continued decline in oak resources. It is important to collect and protect the germplasm resources and ascertain the resource distribution in China. Molecular studies to evaluate the germplasm resources of oaks at the molecular level have been performed. To date, the nuclear genomes of eight species of *Quercus* have been sequenced (*Q. lobata* [109], *Q. robur* [4], *Q. suber* [106], *Q. mongolica* [19], *Q. gilva*, *Q. wutaishanica*, *Q. wislizeni*, *Q. glauca*, and *Q. aquifolioides* (unpublished)), and the chloroplast genomes of 25 species of *Quercus* have been analyzed [110]. This provides basic genetic information for diverse studies on oaks, including studies on their classification, evolution, and breeding. These accumulated findings provide a valuable reference for the evaluation, protection, and effective utilization of oak resources [107].

### 4.2. Promoting the Comprehensive Utilization of Oaks

Evaluating the functional, nutritional, and medicinal value of oaks would provide a reference for their effective utilization. Landscaping projects involving the use of oak trees should consider their ecological adaptability and ornamental properties. Therefore, it is necessary to evaluate the characteristics of different oak tree species, including their stress tolerance (such as tolerance to cold, water, moisture, drought, and salinity), and rapid propagation. The application of oak leaves as a feedstuff for tussahs should focus on improving their quality and yield; therefore, it is important to identify species suitable for the production and growth in different stages of tussah rearing [111]. To effectively utilize oak trees for soil and water conservation, sand fixation, and as wind breaks, it is necessary to identify fast-growing species with high stress tolerance, good adaptability, and excellent performance in soil and water conservation [112].

### 4.3. Studies on Species Introduction, Domestication, Breeding, and Propagation of Oak Trees

Broad-leaved forests dominated by oaks will be gradually replaced by conifer forests, due to fire, drought, or fell threats, such as for *Q. ilex* and *Q. suber* [113], resulting in environmental degradation and severe soil erosion. Moreover, transplanting oak trees from mountains and plains into cities can destabilize the phytocommunity of forest areas. In recent years, China has predominantly used foreign-introduced oaks with differently colored leaves, but has neglected the selection and breeding of indigenous species. This has resulted in the hybridization of oak species. In addition, biotechnologies (such as in genetic engineering) should be used to select and breed new and superior varieties of oaks that are suitable for afforestation, insect feeding, timber production, landscaping, and ecological conservation. Some oak species have undergone transgene editing, such as for herbicide resistance or to enhance tolerance to *Phytophthora cinnamomi* through CsTL1 gene editing a thaumatin-like protein [114,115]. Species introduction and identification should be executed in a planned manner.

### 4.4. Promoting the Utilization of Oak Trees in Urban and Rural Landscaping

Drought and water shortages are the primary constraints on urban ecological landscaping in northern China. Planting drought-tolerant ornamental trees is one of the most effective means of water conservation. Oak trees are preferred as water-conserving and drought-tolerant ornamental trees. In addition, oak trees are of great ornamental value because of their graceful posture, long life, and color variation during autumn. Some oak tree species (such as *Q. texana* and *Q. nigra*) exhibit strong tolerance to salinity stress and are suitable for growth in seasonally flooded low-humidity bottomlands [62]. Owing to its evergreen nature, large canopy, and high tolerance to wind and heavy metals, *Q. virginiana* can be used for the greening of mining wastelands [116]. Thick bark, cork tissues, and leathery leaves are present in *Q. variabilis*, *Q. acutissima*, and *Q. mongolica*. These species are suitable for firebreak forest construction due to their fire and smoke tolerance. After a fire, they will germinate and regenerate quickly. In industrial and mining areas, smoke and harmful gases can be absorbed by *Q*. *brantii* and *Q. mongolica* [117].

### 4.5. Developing Ecological Rearing of Tussah in Oak Forests

To improve the yield of oak leaves and the ecological conditions of oak forests, it is necessary to improve the soil and water conservation and manage fertilization and breeding strategies. In addition, it is advisable to plant oak tree species that are suited to local ecological conditions. Which kinds of oaks are suitable for tussah rearing? How should suitable oak species for tussah varieties in different larvae stages be chosen [111]? Further investigations are needed. Ecological breeding (a balanced breeding objective to maximize the rearing of insects, whilst not impacting the productivity and fitness of the host tree) and oak forest management are both important for tussah rearing.

### 4.6. Improving the Utilization of the Pruning Material of Oak Trees

Small-diameter timbers of oaks can be used as mushroom sticks, and their smashed branches can be used as a substrate and filling material. They are an ideal raw material for cultivating edible mushrooms, such as shii-take and *Ganoderma lucidum*, which are often found under wild oaks. Oak forests of *Q. acutissima* have been used for artificial *G. lucidum* cultivation in China. Other valuable mushrooms, such as porcini, black truffle, and white truffle (also known as “white diamond”), are also grown under oak trees [118,119].

## 5. Conclusions

Oaks play an important role in structuring forest communities, which are a source of wood, firewood, charcoal, and food. They are widely distributed throughout the world and play roles in carbon sinks, greening, soil conservation, energy development, and tussah rearing. In the modern times, oak germplasm resources have been destroyed in some regions of China by excessive utilization in the timber and edible mushroom industries, and as landscaping transplants. At present, more systemic research into existing germplasm resources, plant cultivation, and economic application is needed. Different natural variabilities in resistance to environmental conditions or pathogens from a phenotypic characteristic or genome level should be investigated. Future studies on oaks should focus on the breeding of new varieties, rapid propagation of seedlings, chemical extract investigation, and comprehensive utilization. Molecular technologies, including transgenic technology, gene editing, and tissue culture, should be used in future research. Fully realized oak resource utilization would encompass their ecological, dietary, medicinal, landscaping, and cultural value.

## Figures and Tables

**Figure 1 biology-12-00076-f001:**
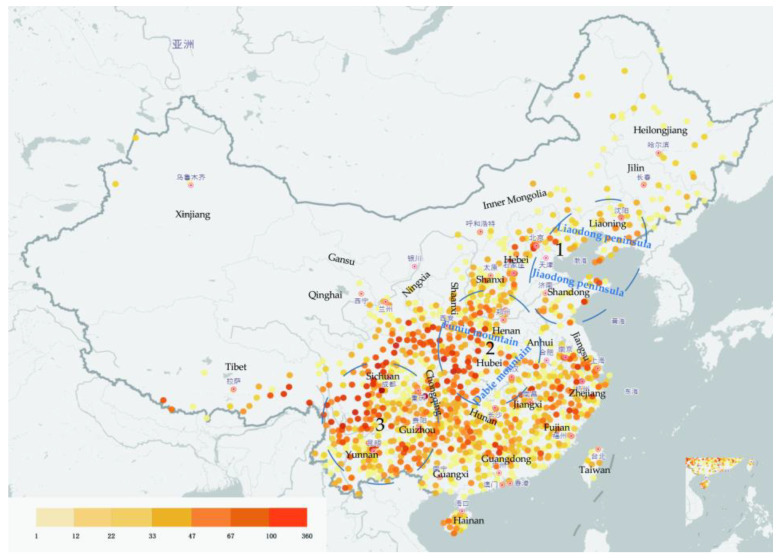
Oak distribution in China based on occurrence records obtained from the National Specimen Information Infrastructure database (www.nsii.org.cn, accessed on 23 December 2022). The different colors represent the collected plant specimen numbers (this reflects the abundance of *Quercus* samples in this area). Tree concentrated distributions in general areas are also shown in the picture. Two *Quercus* species, *Quercus acutissima* and *Quercus dentata* specimen, have been calculated for the top 20 provinces in a Appendix A.

**Figure 2 biology-12-00076-f002:**
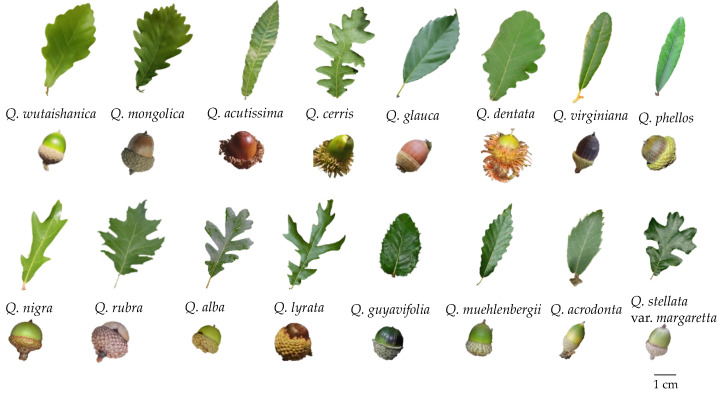
Morphology and diversity of leaves and acorns among a few oak species in China.

**Figure 3 biology-12-00076-f003:**
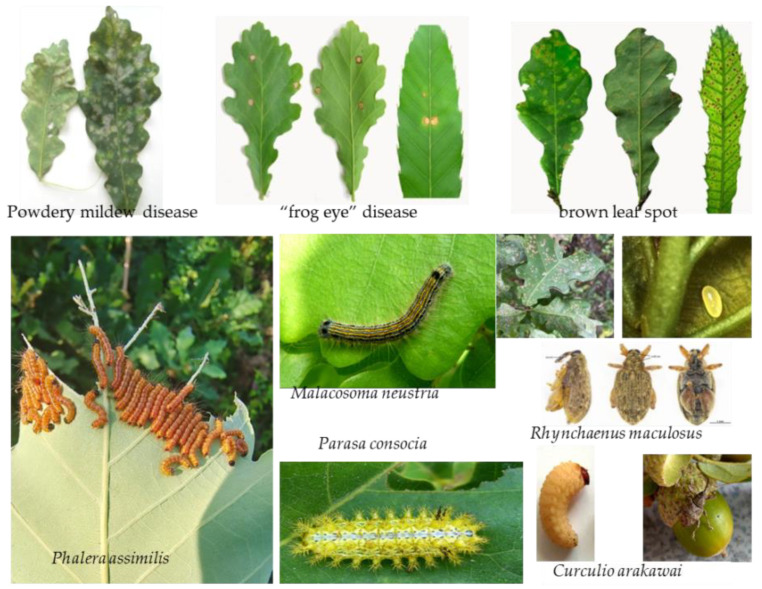
Partial leaf diseases and oak tree pests.

**Figure 4 biology-12-00076-f004:**
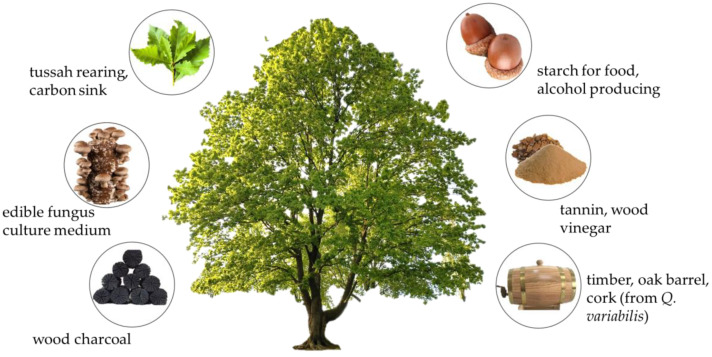
Various commercial products of oak trees in China.

## Data Availability

Not applicable.

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
