# Peer review of "Germplasm Resources of Oaks (Quercus L.) in China: Utilization and Prospects"

_biology, 2022, doi:10.3390/biology12010076_

Round 1
Reviewer 1 Report (Previous Reviewer 1)
I followed the changes introduced by the Authors in the text of the manuscript entitled Germplasm Resources of Oaks (Quercus L.) in China: Utilization and Prospects.
Unfortunately, it is difficult for me to say unequivocally whether the text is suitable for the journal Biology. It seems to me that the Journal's reputation is high today, which is why popular science articles are not serious enough to be presented here. For this reason, I would be inclined to reject it. On the other hand, I believe that by giving the authors a chance to publish (after a thorough reconstruction of the content, quoting and analyzing more data and drawing strictly scientific conclusions based on them), one can contribute to the dissemination of knowledge about an important botanical group. So I would suggest that the Authors try to look at the topic from a more scientific side and resubmit their manuscript in a new form.
Author Response
Dear review, based on your suggestions, we have revised the manuscript in many places. Including the structure, we renew the part 2, make it a more logical structure. Correct the species names, and delete some informal or unaccepted names. Add some introducing about climate influences. I think this review will give reader more details about oak resources distribution and utilization in China. But you have not given any detail considerations, we have modified based on other reviews comments.

Reviewer 2 Report (Previous Reviewer 3)
I thank the authors in addressing the concerns I raised on the manuscript. I believe the edits that have made has greatly improved the readability of the manuscript. I still have a few additional comments that are generally minor in nature but suggest they be addressed before the manuscript is accepted for publication. The line numbers below refer to those in the PDF of corrections with tracked changes.
Abstract
The abstract is still missing where this review points to future research. It only needs to be a sentence that outline where gaps in knowledge remain and need to be the focus of future research.
L27: not sure how ‘un-efficient utilization’ can lead to the decline in oak resources. Do you mean the resources have not been optimally utilized and results in waste. The context doesn’t seem to fit when in the same sentence deforestation and fire are used, which would indeed result in reduced oak resources.
L28-29: is the threatened species listing due to the processes of deforestation, fire, climate change, or combination? Just needs slight clarification.
L29-30: the last point “treats in oak growing” could be rephrased as “threats facing the propagation and growth of oaks in a changing world” or something along those lines.
Introduction
L81: round the percent to whole number (e.g., 10%)
L87-88: not sure what a ‘partial provinces’ is.
L92: add “in China” after “utilization” as the information pertained in the review is of general interest to oaks in China.
Germplasm resources and distribution of oaks
L151-158: is it possible to draw these three main regions on Figure 1?
L170: when mentioning geological epochs, it is good to also include the period. For example, Miocene (3-25 mya). Please check and update throughout.
L298-299: Please add a reference here
L301: could replace “are always influence” with “greatly impact”
L301-303: Could provide more detail on the damage caused. Is it just defoliation or does it cause death? If defoliation, what are the long-term impacts of this browsing on the fitness of these trees? What caused the increase in browsing pressures – e.g., the pest species are managing to get extra breeding cycles due to indirect effects of climate change, like seen in the pine beetle across North America.
L303-304: by itself, this sentence isn’t adding anything. Please add what was the interesting finding.
Utilization of oak trees
I think it would be great to note that the application of the chemical extracts still requires research – this is a key knowledge gap. This was noted by the authors in their response, but it would be important to highlight here and then re-address in the conclusions and next steps forward.
Strategies for the utilization of oaks
L472-473: Please add a reference.
L482-484: Please replace ‘done’ with ‘undergone’. Please replace ‘gene coding’ with ‘gene editing’. Have these GM oaks been used in a commercial setting? Is there concern with the GM varieties escaping and crossing with wild populations? Could this cause flow-on effects to biodiversity? I think answering these will help frame this and the need for risk assessment frameworks - see the gene editing section of Breed et al 2019 which highlights some of these concerns with gene drives etc.
Breed M. F., Harrison P. A., Blyth C., Byrne M., Gaget V., Gellie N. J. C., Groom S. V. C., Hodgson R., Mills J. G., Prowse T. A. A., Steane D. A. and Mohr J. J. (2019) The potential of genomics for restoring ecosystems and biodiversity. Nat Rev Genet 20, 615-628.
L507-509: I'm still not clear on ‘ecological breeding’ and I can not say I have come across this term in the literature. I presume from the edits by the authors, this term refers to a balanced breeding objective to maximize the rearing of insects whilst not impacting the productivity and fitness of the host tree? If this is the case, then something like this could be added to help the reader.
Conclusion
One aspect that is missing from the conclusion is the effect of climate change on oak species in China and the need to better understand the eco-evolutionary processes that have shaped species gene pools. This is a very topical issue and not mentioned throughout the manuscript which I suspect would be a key issue facing oak germplasm resources. Its fine if there is not much reported work in this space, in which case would be the key take home message that these is a massive knowledge gap on how climate change will impact the ecological and economic value of oaks.
Tables/Figures
Table 1: incorporate the table footnote into the table caption.
Figure 1: I wonder if the colouring would be more meaningful if it was based on species richness in a 1km grid? Also the caption could read: “Oak distribution in China based on occurrence records obtained from the National Specimen Information Infrastructure database (website, date accessed). Then detail what the different colours mean – currently the different colours representing different plant specimen numbers has no meaning to a general reader. For example, what does 200 mean? There are 200 occurrences of a species at that given location?
Figure 2: not sure what is mean by a “partial oak species”. Do you mean varieties or hybrids?
Author Response
Thank you for your kindly suggestions. We think the revised manuscript has made a progress follow the comments you have made.

Reviewer 3 Report (New Reviewer)
I think that the list of Quercus species in Table 1 is the part that requires further revision. There is a need to carry out a taxonomic verification to obtain the definitive list of species accepted by the international community. Below I make details of some of the species that require revision according to a conventional source on nomenclatural updates: POWO (https://powo.science.kew.org/)
Another general aspect is the general structure of the document. After the introduction, I suggest that a more logical structure of sections could be: 1) Historical distribution of the species in China, 2) current distribution, 3) Ecological aspects, 4) Use and management of species.
I also observe that in its current format, there is repeated information in some subheadings (for example, in the ancestral uses of some species). In the same way, I find it a bit confusing to distinguish when it is general information about oaks from when they refer to the Chinese oak species.
Specific comments
Line 44. Replace the word “frigid” with “cold”
Lines 57-58 Please modify the original sentence as: A decrease in phosphorus (P) availability has been found under global change [16].
Line 96. Provide a bibliographic reference to support the statement "There are 450 species of oaks in the world."
Lines 147-148, I don't understand, what do you mean when you say “Deciduous oaks are dominant and constructive species…? Please clarify.
Line 263, Consider replacing repopulation instead of “rejuvenation”
Line 326, You mean the names of the taxonomists who have named the Chinese oak species, right? If so, modify the text to make it explicit.
Line 368; Consider replacing title as 3.3. Medicinal value and chemical extracts of oaks
Line 519: Check expression “Oak trees play an important role in building forests.” I think it can be more explicit and direct if the following is said: “Oaks play an important role in structuring forest communities that are a source of wood, firewood, charcoal and food.”
Line 446 onwards. It is necessary to review all references to take care of consistency in writing. For example, there are several cases in the use of lowercase when you must start with uppercase; also several scientific names must be written according to scientific notation.
Table 1. It is necessary to review the taxonomic validity of several taxa and then arrange the names in alphabetical order. In addition, it is convenient that the list of species be presented by section, e.g. Lobatae, Protobalanus, Quercus, and so on.
I recommend adhering to the nomenclature proposed by the Plants of the World Online page (https://powo.science.kew.org/) and adapting the classification here. For example, I found several inconsistencies for the following cases:
1. The correct spelling name to Quercus fabri should be Quercus fabrei. In addition, change Quercus falcate to Quercus falcata Michx., Quercus stellate should be Quercus stellata; Quercus schuette should be Quercus × schuettei Trel.
2. Species not included as valid names in POWO are Quercus fangshanensis Liou, Quercus fenchengensis H. W. Jen et L. M. Wang, Quercus hopeiensis Liou, Quercus kunyuensis, Quercus mongolico-dentata Nakai, Quercus phillyraeoides A. Gray, Quercus serrata Thunb., Quercus aliena var. acutiserrata Maxim. ex Wenz., Quercus aliena var. pekingensis f. jeholensis (Liou et Li), Quercus stellata var. margaretta.
3. Quercus liaotungensis Koidz. is a synonym of Quercus mongolica var. crispula (Blume) H.Ohashi.
4. Quercus lanceolata S.Z.Qu & W.H.Zhang is a synonym of Quercus shangxiensis Z.K.Zhou.
5. Quercus malacotricha A.Camus is a synonym of Quercus dentata subsp. yunnanensis (Franch.) Menitsky)
6. Quercus stewardii Rehder is a synonym of Quercus dentata subsp. stewardii (Rehder) A.Camus
7. Quercus acutissima var. septentrionalis Liou is a synonym of Quercus acutissima subsp. acutissima Carruth
8. Quercus acutissima var. depressinucata H.Wei Jen & R.Q.Gao is a synonym of Quercus acutissima subsp. acutissima
9. Quercus baronii var. capillata (Kozlov) Liou is a synonym of Quercus baronii Skan
10. Quercus dolicholepis var. elliptica (Y.C.Hsu & H.Wei Jen) Y.C.Hsu & H.Wei Jen is a synonym of Quercus dolicholepis A.Camus
11. Quercus mongolica var. grosseserrata (Blume) Rehder & E.H.Wilson is a synonym of Quercus mongolica var. crispula
12. Quercus mongolica var. macrocarpa H.Wei Jen & L.M.Wang is a synonym of Quercus mongolica var. mongolica
13. Quercus glandulifera var. stellatopilosa W.H.Zhang is a synonym of Quercus serrata Murray
14. Quercus serrata var. brevipetiolata (A.DC.) Nakai is a synonym of Quercus serrata Murray
15. Quercus serrata var. tomentosa (B.C.Ding & T.B.Chao) Y.C.Hsu & H.Wei Jen is a synonym of Quercus serrata Murray
16. Quercus variabilis var. pyramidalis T.B.Chao, Z.I.Chang & W.C.Li is a synonym of Quercus variabilis Blume
17. Quercus macrolapsis is misspelled. Could it be Quercus macrolepis Kotschy?
Author Response
Thank you for the review work. Especially the details about the species name correction. We have carefully check every name of the oak species.

Round 2
Reviewer 1 Report (Previous Reviewer 1)
I have no comments.
This manuscript is a resubmission of an earlier submission. The following is a list of the peer review reports and author responses from that submission.
Round 1
Reviewer 1 Report
Dear Authors,
The Review article provides interesting information on a group of plants that is and has been an extremely important part of the Chinese flora. The text is written in an accessible language and its layout seems to be correct.
However, I would like to point out what, in my opinion, is missing from the manuscript.
On the merits:
1. The distribution of Quercus species is described in a minimalist way. In this case, the characteristics of at least selected species (especially numerous or extremely rare) should be presented in more detail (with data on population resources), especially that we are dealing here with both native species and species imported to China in the past.
2. Here a question arises about the ecological status of populations of various oak species (including introduced ones), there is no detailed information on the threats to their functioning (natural and anthropogenic), and information on whether they include species that are under protection or should be protected. Maybe at least some information is already available?
Detailed suggestions:
Lines 110-113 the subject needs to be developed, because there is data on the conducted research, it should be mentioned and summarized for the readers
Lines 117-118 similarly to the above, it is interesting if we know what was the fate of the introduced species, are they still present in China, or was the introduction completely unsuccessful?
Tab. 1 It seems to me that it is possible to separate native species from introduced species. Citations should be placed in the table caption or in footnotes below the table. If data on the population status of these species are known, they can be grouped into those with numerous populations and those with limited resources.
Fig. 1 and fig 3 no citation, was the drawing made by the authors of the manuscript?
Line 230-231 no citations
Line 243-247 no citations
Line 251-264 no citations
Line 268-271 no citations
Lines 345-346 no citations
Lines 352-356 no citations
Lines 370-373 no citations
Lines 375-384 no citations
Lines 406-411 no citations
Line 412-419 these are conclusions rather than overview information
‘Conclusions’ contain information that has not been analysed in manuscript, e.g. ... Oaks play important roles of afforestation of barren mountains, soil conservation, water retention ...
The manuscript also does not contain data on populations resource destruction, so the conclusions here are also not the result of an analysis carried out on the basis of the source data.
Authors should remember, above all, that in a Review type scientific article, all information presented in it must be supported by appropriate literature sources.
I believe that the article should be expanded with the missing issues so that it could be a valuable source of data for other scientists.
Reviewer 2 Report
After reading the submitted review paper entitled "Germplasm Resources of Oaks (Quercus L.) in China: Utilization and Prospects", I have mixed feelings about whether it should be considered for publication in a scientific journal. On the one hand, this paper summarizes much information about oak species growing in China. On the other hand, one could ask what new such a review brings and whether it is needed. The authors should justify this in the introduction. Evaluation of the manuscript is hampered by poor English, which makes many sentences sound strange, incomprehensible or incorrect. For example, the authors wrote, "Here, we review the ancient books, fossil discoveries, and modern basic research on oaks." However, the oldest cited publication in the literature list is from 1983. In my part of the world, antiquity ended more than 1500 years ago, if I remember correctly. Hence, I don't know what the authors mean when they write about ancient books.
I can only add comments to the text when it has been corrected. I recommend that the authors seek help from someone experienced in writing scientific botanical publications. A translator without experience with such texts will not cope and may further distort the meaning of sentences.
It is unclear why they included Table 1 with a list of species in the main text. I propose to remove it. I also don't know what the purpose of Figure 1 is. The sentence quoting it adds nothing of substance. The authors should add explanations for maps in Figure 2. Why are two maps shown instead of one? Do the colours indicate the number of species or something else?
In conclusion, the authors wrote, "Future studies on oaks should focus on breeding of new varieties, rapid propagation of seedlings, and comprehensive utilization."
The authors immediately suggest tampering with oak genomes. Shouldn't their natural variability be looked at first? What about comparative studies of species? Sometimes it is enough to study and understand the natural variability of plant populations to find suitable populations that are resistant to unfavourable environmental conditions, pathogens or have greater utility in specific materials production.
In my opinion, the problem of climate change has been completely ignored here. Oaks are long-growing trees, and climate change is occurring fast enough that young trees may not reach maturity. Droughts and late frosts are an increasing threat to oaks and other tree species in seasonally variable climates, and this is due to climate change. Certainly, the problem also affects the China area. I think it is worth suggesting similar studies that have recently been conducted on oaks in Europe (Čehulić et al 2019) and on spruces in North America (Gio et al. 2020).
Čehulić, I., Sever, K., Katičić Bogdan, I., Jazbec, A., Škvorc, Ž., Bogdan, S., 2019. Drought Impact on Leaf Phenology and Spring Frost Susceptibility in a Quercus robur L. Provenance Trial. Forests 10, f10010050. https://doi.org/10.3390/f10010050
Guo, X., Klisz, M., Puchałka, R., Silvestro, R., Faubert, P., Belien, E., Huang, J., Rossi, S., 2022. Common‐garden experiment reveals clinal trends of bud phenology in black spruce populations from a latitudinal gradient in the boreal forest. J. Ecol. 110, 1043–1053. https://doi.org/10.1111/1365-2745.13582
Puchałka, R., Koprowski, M., Przybylak, J., Przybylak, R., Dąbrowski, H.P., 2016. Did the late spring frost in 2007 and 2011 affect tree-ring width and earlywood vessel size in Pedunculate oak (Quercus robur) in northern Poland? Int. J. Biometeorol. 60, 1143–1150. https://doi.org/10.1007/s00484-015-1107-6
Puchałka, R., Koprowski, M., Gričar, J., Przybylak, R., 2017. Does tree-ring formation follow leaf phenology in Pedunculate oak (Quercus robur L.)? Eur. J. For. Res. 136, 259–268. https://doi.org/10.1007/s10342-017-1026-7
Sohar, K., Helama, S., Läänelaid, A., Raisio, J., Tuomenvirta, H., 2013. Oak decline in a southern Finnish forest as affected by a drought sequence. Geochronometria 41, 92–103. https://doi.org/10.2478/s13386-013-0137-2
Doležal, J., Lehečkova, E., Sohar, K., Altman, J., 2016. Oak decline induced by mistletoe , competition and climate change : a case study from central Europe. Preslia 88, 323–346.
I will add detailed comments to the text if the authors decide to make language corrections. Currently, the manuscript is chaotic and not easy to understand in parts.
Reviewer 3 Report
Summary
This manuscript reviews the distribution, history, and utilization of oak species in China. The authors highlight the cultural and commercial importance of this group of diverse species and point to future research directions. While the authors have reviewed a reasonable body of literature, there are many areas throughout the manuscript where literature should have been cited to support the arguments made or when summary statistics (for example, percentages) are provided. Further, the scope of the literature review was not clear, particularly there was a lack of objectives that the review hoped to achieve. For this reason, I can not adequately comment on where key literature is missing and thus only provide high-level comments that might help the authors clarify and refine the review. I suggest the authors look at other reviews on the oaks (such as https://doi.org/10.1111/nph.15450). Lastly, while I appreciate that this manuscript focuses on provinces of China, a map could benefit of the key provinces mentioned in the manuscript may benefit international readers who may not be as familiar with the many provinces of China. This could be simply achieved by adding the provinces to Fig 2. I have sub-divided my comments by the major sections of the paper and line numbers refer to the pdf version of the manuscript.
Title
The current title does not match the content that is within the paper. The majority of the paper appears to be a historical account of oaks in China, their distribution, and their economic and culture significance. There is little mention of germplasm resources. Maybe a suggested title is “A historical account of the economic and cultural significance of oaks in China”.
Simple Summary & Abstract
L16: Its not clear what “it” is in the sentence “It will provide a deeper understanding…”.
L18-30: Please consider re-writing the abstract. It should provide a brief background for the study, state the objectives/knowledge gaps that the review hopes to achieve/fill, and a summary of the main take home messages followed by a conclusion that points to future issues or research.
Introduction
L35: Could remove the inverted commas around “Oak trees” as this is unneeded.
L44-46: Please provide a reference(s) here to back up this statement
L46-48: Could read “Oak tree classification is challenging work partly due to inter- or intra-specific hybridization and introgression [5]”.
L48-49: These two sentences could be merged: “Oak leaves vary in size and shape among different species and likely represents an adaptive response to changing environmental conditions [6,7]”.
L50: Could read: “Genomics studies have also investigated patterns of adaptation in…”
L58: Replace “always” with “often”.
L59: After Chuzhou city please added (China) for clarification.
L72-75: The importance of oaks in China is apparent in the second paragraph of the introduction. However, the issues or knowledge gaps that this review hopes to address is not clear. What has motivated the authors to do this review? What are the objectives of the review? What is not reviewed here but captured in other reviews? While some of this is here, it needs further elaboration and hopefully answering some of these informal questions might prompt some text that can be added here to frame the purpose and importance of the literature review.
Germplasm resources and Distribution of Oaks
L93: Replace “…can feed the…” with “provide critical habitat for”.
L94: Replace “70 thousand” with “70,000”. Please check through out and make similar changes (e.g., L96).
L96: Replace “…oak, and only…” with “…oak, with…” and remove “are” in “…tussah cocoons are produced…”.
L103-104: Please unpack this a little more. Why not use oaks that are native to China? Were the introduced oak trees planted beyond their native ranges?
L127-175: This section is broad-scale characteristics of the species. Is there evidence of local adaptation and what environmental factors have shaped this adaptive variation among populations of a species?
L127: Replace “Oak leaves variation in leaf size…” with “Oak vary in their leaf size…”. Also, replace “These variations…” with “This variation…”.
L128: Delete “plant” and replace “and” with “or”.
L128-129: This sentence needs reference(s).
L129-133: These sentences need referencing, unless it is all within [18] in which case this needs to be acknowledged at the start.
L146-147: Please add a reference for this sentence.
L155-158: Is “infertility” referring to the soil, if so, please add “soil infertility”.
L164-166: Please rephrase this sentence as it is not clear what is being raised and unfortunately does not make sense. What is “geographic evolution”? Are the authors stating that evergreen species have populations that are deciduous or do they mean that as the climate changes across elevation evergreen species are replaced by deciduous species? Please also add references to support this.
L166-167: Please clarify what is meant by “evolution is still incomplete”. Also please add references.
L176-241: I suggest moving this subsection to the start of section 2 as it sets the scene for the geographic spread of the various oak species in China, particularly adding the key provinces mentioned in the manuscript so international readers can easily orientate themselves. This section could also benefit by doing a historical account that starts from the fossil records to more recent contemporary times. After this geographic distribution of oaks in China, I would then suggest the next section comprising the environmental spread and how they adapt to their environment.
L185: Replace “Their vertical distribution ranges a few meters to…” with “Their altitudinal distribution ranges from near sea level to…”.
L188: Does “Area” need to be capitalized?
L202: Suggest adding the time period of mentioned geological periods, for example “Miocene (5-23 mya).”.
L225: Please clarify what trees are mentioned when stating “these trees”.
Utilization of Oak Trees
L251-264: This section needs referencing, especially where wood property values are provided.
L257-261: Could compare these wood properties to other commercial hardwood and softwood tree species.
L265: Please revise this subsection title – feed is the verb of food.
L269: Please define the time period that associates with “prehistoric times”
L272: Suggest “The leaves of oak trees” rather than “Oak tree leaves”.
L279: Is “feedstuff” the right word here as it is usually used in a livestock context. Maybe “food source”?
L280-282: This sentence repeats information already provided in section 2.1.1 – suggest deleting.
L306-324: I struggle to understand the point of this paragraph. It provides interesting chemical compositional differences among species, but why is this important in a medical context?
L352-353: Ciuld replace “…however, pollen poplar causes…” with “…however, its pollen can cause…”.
L355: Please clarify what is meant by “highest evaluation” in the context of this sentence.
Strategies for the Utilization of Oaks
L365-366: Delete “biological” from “Molecular biological”. Please clarify what is meant by “evaluate”; do the authors mean genetic diversity? Delete “…at the molecular level…”.
L380-382: Have breeding programs been established to selectively improve the leaves for tussah production? Are the other studied that show there is heritable variation in other oaks to suggest there is potential for breeding programs to improve traits of interest?
L387: What sort of destruction? Fire, wind damage, drought/heat stress?
L389-391: This sentence needs referencing.
L393-397: Would genetically modified variants be a socially and ethically viable conservation option? What would they be superior in, growth/productivity or browsing resistance or climate resilience? Are there precedent studies in other oak species (e.g., American oak) where this has occurred, and are there lessons of success/failure that can be translated and applied to oaks from China?
L409: Delete “a certain”.
L411: Should “gas” be “gases”?
L418-419: I am not sure what is meant by “Ecological breeding technologies”?
Conclusion
L430: Delete “and” in “and energy development”. Also delete the duplicated “are”
Figures & Tables
Figure 1: Is this a reprint from a previous publication? If not, please add a credit for the images.
Figure 2: The figure caption needs a bit of work. What do the different colors represent? Why are the two datasets separated and not plotted on the one map? Could potentially (i) map the spatial pattern of oak richness, and (ii) map the key species that are highlighted in the text as different colors with inset pictures of what the trees look like for the vegetive communities they support.
Figure 3: Caption could read: “Diverse commercial products of oak trees in China”.
Reviewer 4 Report
The paper provides a lot of information that is of an overview nature. Kudos to the authors for a dedicated work that required many reviewed and read literature. The paper will be of importance to the scientific public, especially for making recommendations in accordance with different climate scenarios. I suggest the paper be accepted in its current form and published as a review paper.